# Cross-neutralization of SARS-CoV-2 by HIV-1 specific broadly neutralizing antibodies and polyclonal plasma

**Nitesh Mishra**[1], **Sanjeev Kumar**[1,2], **Swarandeep Singh**[1], **Tanu Bansal**[1], **Nishkarsh Jain**[1], **Sumedha Saluja**[1], **Rajesh Kumar**[3], **Sankar Bhattacharyya**[3], **Jayanth Kumar Palanichamy**[1], **Riyaz Ahmad Mir**[1], **Subrata Sinha**[1], **Kalpana Luthra**[1]*

**1** Department of Biochemistry, All India Institute of Medical Sciences, New Delhi, India, **2** ICGEB-Emory Vaccine Centre Program, International Centre for Genetic Engineering and Biotechnology, New Delhi, India, **3** Translational Health Science and Technology Institute, NCR Biotech Science Cluster, Faridabad, Haryana, India

* kalpanaluthra@gmail.com

**Data Availability Statement:** All relevant data are within the manuscript.

**Funding:** This work was supported in part by Science and Engineering Research Board (SERB),

## Abstract

Cross-reactive epitopes (CREs) are similar epitopes on viruses that are recognized or neutralized by same antibodies. The S protein of SARS-CoV-2, similar to type I fusion proteins of viruses such as HIV-1 envelope (Env) and influenza hemagglutinin, is heavily glycosylated. Viral Env glycans, though host derived, are distinctly processed and thereby recognized or accommodated during antibody responses. In recent years, highly potent and/or broadly neutralizing human monoclonal antibodies (bnAbs) that are generated in chronic HIV-1 infections have been defined. These bnAbs exhibit atypical features such as extensive somatic hypermutations, long complementary determining region (CDR) lengths, tyrosine sulfation and presence of insertions/deletions, enabling them to effectively neutralize diverse HIV-1 viruses despite extensive variations within the core epitopes they recognize. As some of the HIV-1 bnAbs have evolved to recognize the dense viral glycans and cross-reactive epitopes (CREs), we assessed if these bnAbs cross-react with SARS-CoV-2. Several HIV-1 bnAbs showed cross-reactivity with SARS-CoV-2 while one HIV-1 CD4 binding site bnAb, N6, neutralized SARS-CoV-2. Furthermore, neutralizing plasma antibodies of chronically HIV-1 infected children showed cross neutralizing activity against SARS-CoV-2 pseudoviruses. Collectively, our observations suggest that human monoclonal antibodies tolerating extensive epitope variability can be leveraged to neutralize pathogens with related antigenic profile.

## Author summary

In the current ongoing COVID-19 pandemic, neutralizing antibodies have been shown to be a critical feature of recovered patients. HIV-1 bnAbs recognize extensively diverse cross-reactive epitopes and tolerate diversity within their core epitope. Given the unique nature of HIV-1 bnAbs and their ability to recognize and/or accommodate viral glycans,

Department of Science and Technology (DST), India (EMR/2015/001276) and Department of Biotechnology (DBT), India (BT/PR5066/MED/1582/2012). S.K. is supported through DBT/Wellcome Trust India Alliance Early Career Fellowship grant IA/E/18/1/504307. The funders had no role in study design, data collection and analysis, decision to publish, or preparation of the manuscript.

**Competing interests:** The authors have declared that no competing interests exist.

we reasoned that the glycan shield of SARS-CoV-2 spike protein can be targeted by HIV-1 specific bnAbs. Herein, we showed that HIV-1 specific monoclonal antibodies as well as polyclonal plasma antibodies cross-react and neutralize SARS-CoV-2. Understanding cross-reactive neutralization epitopes of antibodies generated in divergent viral infections will provide key evidence for engineering so called super-antibodies (antibodies that can potently neutralize diverse pathogens with similar antigenic features). Such cross-reactive antibodies can provide a blueprint upon which synthetic variants can be generated in the face of future pandemics.

## Introduction

Broadly neutralizing antibodies (bnAbs) targeting the HIV-1 envelope glycoprotein (Env) can neutralize a broad range of circulating HIV-1 isolates and have been called super-antibodies due to their remarkable potency and neutralization breadth against viruses with significant alterations in the core epitope [1]. As a result of its extensive genetic diversity, HIV-1 is subdivided in multiple clades and circulating recombinant forms (CRFs). A rare subset of HIV-1 infected individuals develops broad and potent antibody responses and have served as potential candidates for the isolation of HIV-1 bnAbs [2,3]. HIV-1 bnAbs take years to develop, have atypical features including long complementarity-determining regions (CDR) loops, high levels of somatic hypermutations (SHMs), presence of insertions and/or deletions (indels), tyrosine sulfation, and develop to tolerate significant alterations in their core epitope [1–3]. Notably, V2-apex bnAbs have been shown to exhibit cross-group neutralization activity with viruses derived from HIV-1 group M, N, O and P Envs. Furthermore, they even show cross-neutralization of simian immunodeficiency virus (SIV) isolates [4]. Though such extensive cross-neutralization potential of several bnAbs is often associated with poly and/or autoreactivity. Several glycan-dependent HIV-1 bnAbs, such as PGT121 and PGT151, have been shown to bind uninfected/bystander cells.

Severe acute respiratory syndrome coronavirus 2 (SARS-CoV-2) emerged in late 2019, rapidly spread across different countries, infecting millions of individuals and has caused a global COVID-19 pandemic[5]. The SARS-CoV-2 trimeric spike glycoprotein (S) binds to angiotensin-converting enzyme 2 (ACE2) which leads to host cell entry and fusion [6,7]. Type 1 viral fusion machines, including HIV-1 Env, Influenza hemagglutinin (HA), and SARS-CoV-2 S protein, mediate viral entry driven by structural rearrangements and are trimeric in their prefusion and post-fusion state [2,7,8]. SARS-CoV-2 S protein is covered by host-derived glycans on 66 PNGS on each trimer and site-specific glycan analysis has shown that 28% of glycans on the protein surface are underprocessed oligomannose-type glycans [9]. SARS-CoV and HCoV OC43 elicited antibodies have been shown to cross-react with SARS-CoV-2. The Neutralizing antibody (nAb), S309, isolated from memory B-cells of a SARS-CoV infected individual targets a glycan epitope conserved within the Sarbecovirus subgenus [10]. Several HIV-1 bnAbs have been shown to penetrate the glycan shield and contact protein residues in Env via their long complementary determining region (CDR) loops and make stabilizing contacts with the surrounding high mannose and complex glycans [11,12]. Several HIV-1 bnAbs recognize glycopeptides and/or cluster of N-linked glycans [1–3]. The glycans on HIV-1 Env are highly dynamic and can be occupied by different glycoforms due to glycan processing. The glycan shield covering the HIV-1 Env comprises roughly half its mass and shields approximately 70% of the protein surface with glycosylation occurring on potential N-linked glycosylation sites that vary significantly between infected individuals (18–33 PNGS) [13,14].

Herein, we reasoned that given the unique nature of HIV-1 bnAbs and their ability to recognize and/or accommodate viral glycans, the glycan shield of SARS-CoV-2 spike protein can be targeted by HIV-1 specific bnAbs.

## Results

### Broadly neutralizing antibodies targeting HIV-1 Env cross-react with SARS-CoV-2 spike protein

In the past decade, a large panel of bnAbs and non-nAbs targeting diverse epitopes on the HIV-1 Env glycoprotein (gp160) have been isolated and extensively characterized (reviewed in refs [1–3]). To evaluate the potential cross-reactivity of these antibodies, we first performed binding ELISA of 30 bnAbs and 7 non-nAbs with SARS-CoV-2 S2P$_{ecto}$ protein (pre-fusion stabilized ectodomain construct, 1–1208 amino acid residues) and receptor-binding domain (RBD, residues 319–541, also called S1$^B$ domain). The HIV-1 bnAbs were categorized into five categories based on their epitopes on the HIV-1 viral Env (Fig 1A). CR3022 (a nAb isolated from a convalescent SARS-CoV patient [15], which has been shown to cross-react with SARS-CoV-2[7]), and CC12.1 (a SARS-CoV-2 neutralizing antibody isolated from a convalescent donor) were used as positive control while two antibodies targeting the envelope glycoprotein of simian immunodeficiency virus (SIV) were used as negative control. Of the 30 HIV-1 bnAbs tested for binding to both S2P$_{ecto}$ protein and RBD of SARS-CoV-2, 6 bnAbs (VRC07.523LS, NIH45-46 G54W, N6, Z13e1, 2F5 and 4E10) showed significant binding, while one bnAb (CAP256.09) showed moderate binding to only the S2P$_{ecto}$ protein (Fig 1A). Non-nAbs (non-neutralizing antibodies) that target post-fusion and/or open trimeric conformation of HIV-1 Env were unable to bind both SARS-CoV-2 S2P$_{ecto}$ protein and RBD protein (Fig 1B), suggesting that only pre-fusion state specific antibodies that evolve via extensive somatic hypermutation and affinity maturation in response to repeated exposure to a continuously evolving antigen can cross react with other viruses.

Though reactivity against RBD was stronger than S2P$_{ecto}$ protein for majority of the bnAbs tested, a significant positive correlation was seen between binding to S2P$_{ecto}$ protein and RBD (Fig 1C). All the 6 bnAbs that showed binding reactivity in ELISA exhibited a similar binding profile to the cell surface expressed SARS-CoV-2 S glycoprotein (Fig 1D). Of the 30 monoclonal antibodies tested herein, bnAbs targeting the membrane proximal external region (MPER) of HIV-1 showed maximum binding to both the S protein and RBD with half-maximal effective concentration (EC$_{50}$) of 0.71 μg/ml and 1.71 μg/ml (Z13e1), 0.048 μg/ml and 2.91 μg/ml (2F5) and 0.79 and 0.33 μg/ml (4E10) μg/ml to the RBD and S2P$_{ecto}$ respectively. VRC07.523LS is an engineered variant of the VRC01 bnAb with higher SHM [16] and while it showed binding to both S2P$_{ecto}$ protein and RBD, both VRC01 and its somatic related clone VRC03 did not show any binding (Fig 2A and 2B) further suggesting that higher SHM in HIV-1 bnAbs might be responsible for the observed cross-reactivity.

### Optimized conditions for neutralization of pseudotyped coronaviruses

We next tested the ability of all six bnAbs that showed binding to SARS-CoV-2 spike ectodomain and RBD to block SARS-CoV-2 infection in a lentiviral (HIV-1) based pseudovirus neutralization assay[17–20]. We first generated HIV-1 lentiviral particles pseudotyped with full-length spike proteins from SARS-CoV-2 using a replication-deficient HIV-1 proviral plasmid (pNL4-3.Luc.R-E-). Pseudovirus infection was tested in HEK293T, HEK293T/ACE2 cells (HEK293T cells stably transduced with human ACE2), Vero-E6 cells, and Huh7 cells. Both Vero-E6 and Huh7 cells endogenously express ACE2 and have been shown to be susceptible

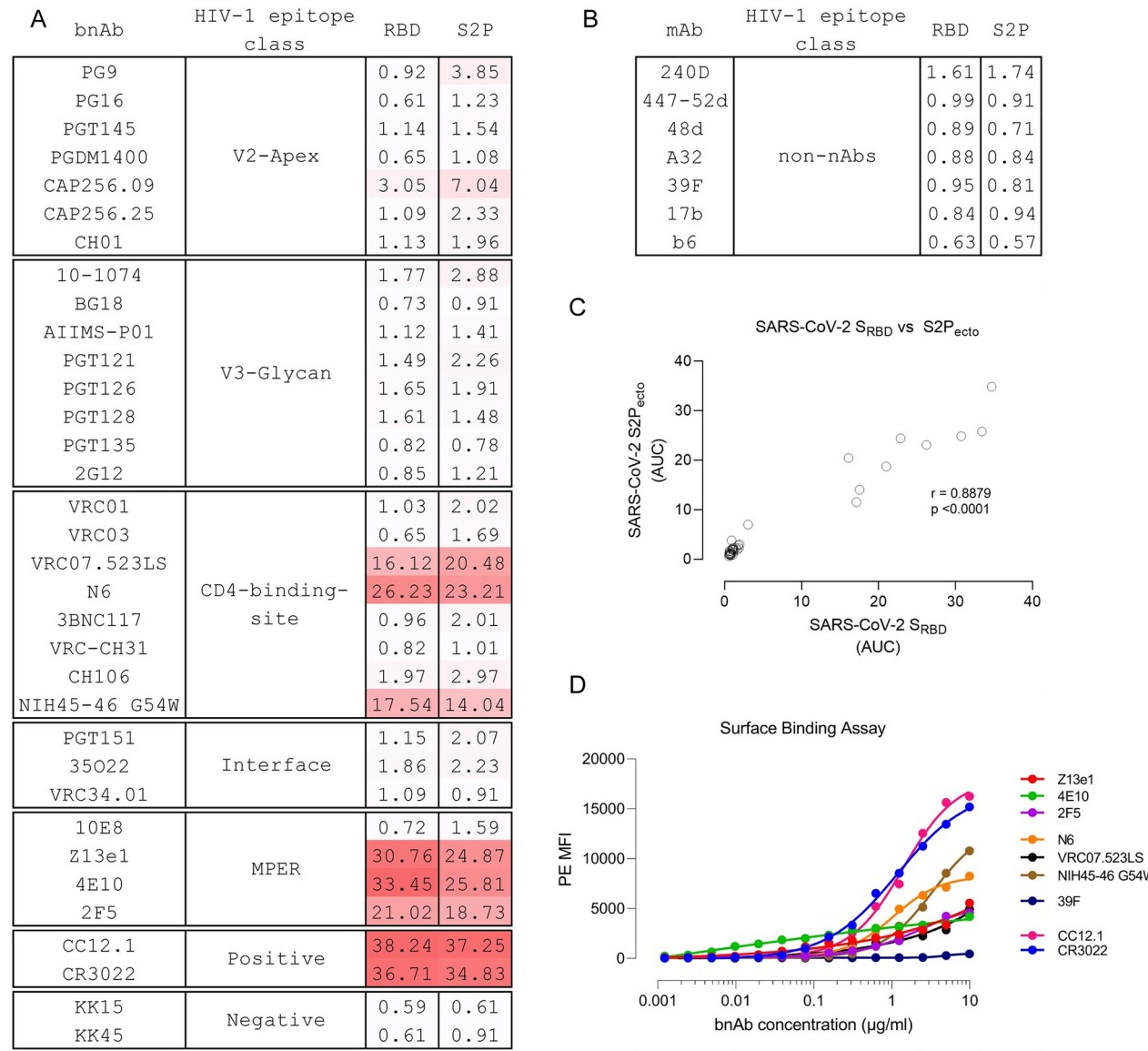

**Fig 1. HIV-1 bnAbs cross-react with the receptor binding domain of SARS-CoV-2.** (A–B) Cross-reactivity of anti-HIV-1 broadly neutralizing antibodies targeting diverse epitopes on HIV-1 Env and non-neutralizing antibodies were assessed by ELISA using SARS-CoV-2$_{RBD}$ and SARS-CoV-2 S2P$_{ecto}$. CR3022, a SARS-CoV neutralizing antibody, was used as positive control. Two antibodies targeting SIV Env were used as negative control. Area under curve (AUC) of OD$_{450}$ values of a 12-point binding curve (range, 0.0048 to 10 μg/ml) from three independent experiments are shown. (C) Two-tailed Spearman's correlation was calculated using the area under curve (AUC) values. A significant positive correlation was observed between RBD and S2P$_{ecto}$ (spearman r = 0.8879, p <0.0001). (D) Binding of HIV-1 bnAbs that showed cross-reactivity to S2P and RBD domain of SARS-CoV-2 in ELISA to full-length SARS-CoV-2 S glycoprotein expressed on the surface of HEK293T cells. Average median fluorescence intensity values of a 12-point binding curve (range, 0.0048 to 10 μg/ml) from three independent experiments were used to draw the curve. CR3022, a SARS-CoV neutralizing antibody, and CC12.1, a SARS-CoV-2 neutralizing antibody, were used as positive control.

to infection by all seven coronaviruses [21–23]. No infection of HEK293T cells with SARS-CoV-2 pseudoviruses was observed (Fig 3A and 3D). HEK293T/ACE2 cells and Vero-E6 cells showed comparable infection though the infection titres typically remained in the range of 1 x $10^4$ to 2 x $10^5$ while Huh7 cells were moderately infected with infection titre in the range of $10^3$ to $10^4$ (Fig 3A).

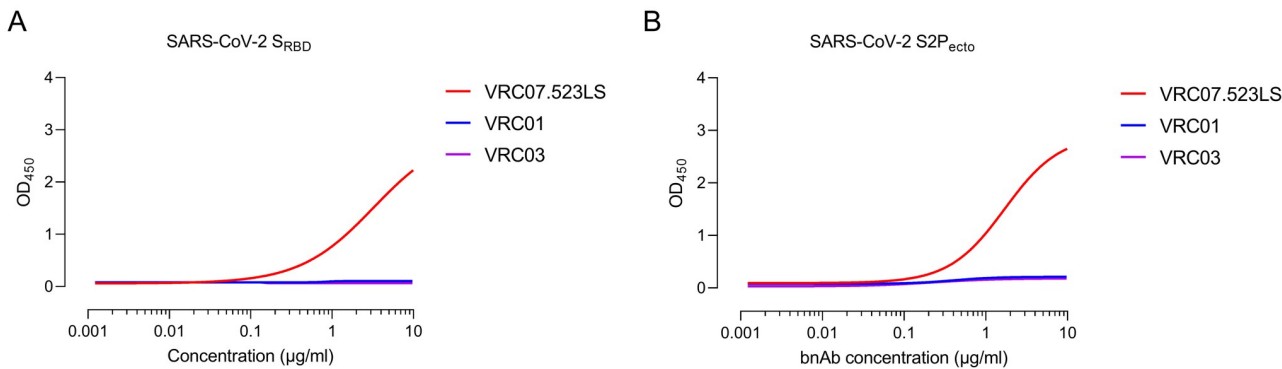

**Fig 2. Somatically engineered VRC07.523LS cross-reacts with SARS-CoV-2.** Cross-reactivity of anti-HIV-1 broadly neutralizing antibodies, VRC07.523LS, VRC01 and VRC03, targeting the CD4-binding site on HIV-1 Env. Cross-reactivity was assessed by ELISA using (a) SARS-CoV-2$_{RBD}$ and (b) SARS-CoV-2 S2P$_{ecto}$. OD$_{450}$, optical density at 450 nm. OD$_{450}$ values are from a 12-point binding curve (range, 0.0048 to 10 µg/ml).

Truncation of the SARS-CoV-2 spike has been shown to enhance production of viral pseudoviruses [19,20,24,25]. We next utilized a truncated version of SARS-CoV-2 lacking the last 19 amino acids from the C terminal (Δ19). Compared to wildtype spike, Δ19 spike showed 2-3-fold increase in infection titre in all three cell lines (Fig 3B). Since the HIV-1 proviral plasmid used (pNL4-3.Luc.R-E-) contains luciferase gene in the same frame as HIV-1 Env, we reasoned that using a separate plasmid containing luciferase gene under a strong constitutive promoter (CMV promoter) can further increase the pseudoviral titres. Next, we generated pseudoviruses using a three-plasmid system (HIV-1 proviral backbone lacking Env and luciferase, a CMV-luciferase construct with HIV-1 packaging signal and Δ19 spike protein). Compared to two-plasmid system with wildtype SARS-CoV-2 spike, three-plasmid system with

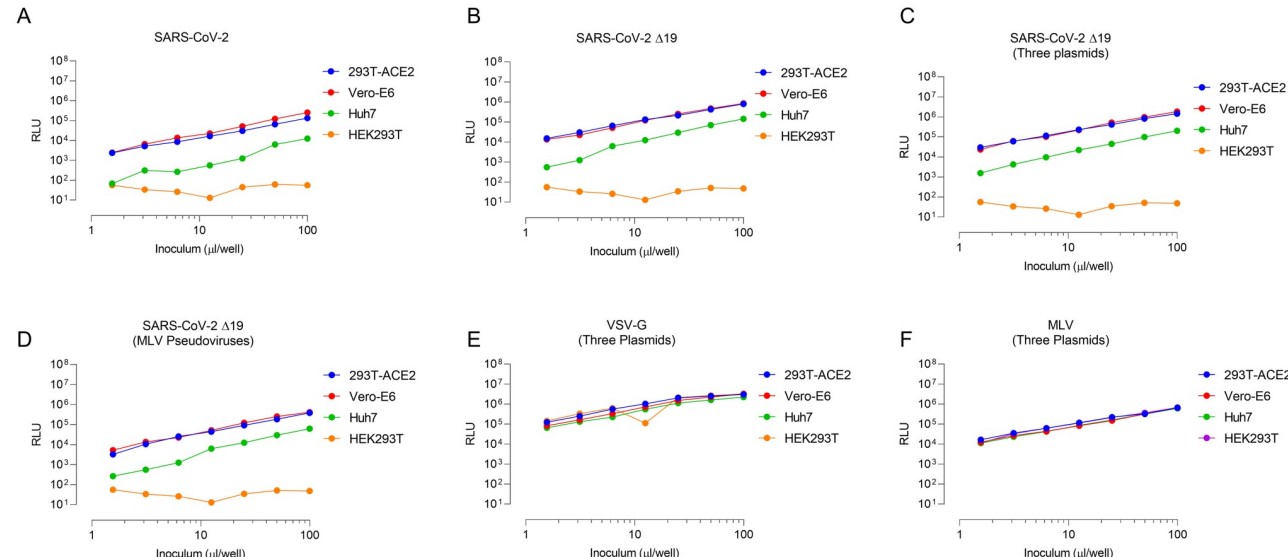

**Fig 3. Lentiviral (HIV-1) particles pseudotyped with SARS-CoV-2 spike productively infect 293T/ACE2 and Vero-E6 cells.** (A–D) Infectivity measurements of SARS-CoV-2 pseudoviruses on the indicated cell lines. Infectivity was quantified by measuring luciferase activity (relative light units, RLU) following infection of cells in 96-well plates with the indicated volume (inoculum, µl/well) of pseudotyped viruses. Pseudoviruses were generated by using two plasmid system (HIV-1 proviral backbone containing luciferase gene and SARS-CoV-2 spike) or three plasmid system (HIV-1 proviral backbone, Luciferase reporter plasmid and SARS-CoV-2 spike). Infectivity assays were performed twice in triplicates and average RLU are shown. (E–F) Amphoteric VSV-G and MLV pseudoviruses were used as positive virus infection controls.

Δ19 spike, on average, provided ~10-fold higher titres (Fig 3C). To increase the robustness of neutralization assays, we also employed a MLV particles-based neutralization assay using a MLV proviral backbone, CMV-luciferase vector with MLV packaging signal and Δ19 spike protein. Compared to HIV-1 based pseudoviruses, MLV pseudoviruses typically showed low titres (~2-3-fold lower titres) (Fig 3D). For all infection studies, amphotropic envelope protein from Vesicular Stomatitis Virus (VSV-G) and murine leukemia virus (MLV) were used as positive infection controls (Fig 3E and 3F).

We, therefore, employed the three plasmid HIV-1 pseudovirus assay with HEK293T-ACE2 cells as they also provided as to use HEK293T cells as negative infection controls.

## N6, a bnAb targeting HIV-1 CD4bs, neutralized SARS-CoV-2 pseudoviruses but failed to neutralize authentic virus

All 6 HIV-1 bnAbs that showed binding to SARS-CoV-2 S protein and RBD were then tested for their ability to block infection using a HIV-1 pseudovirus based neutralization assay utilizing SARS-CoV-2 spike protein. VSV-G and MLV pseudotyped viruses were used as negative control. Except N6, all remaining five bnAbs failed to neutralize SARS-CoV-2 (Fig 4A). Though N6 showed neutralization of SARS-CoV-2, it failed to show complete neutralization (maximum percent neutralization of 88% with an $IC_{50}$ of 0.988 μg/ml) and compared to an anti-RBD nAb isolated from a convalescent donor, CC12.1, N6 showed ~100-fold weak neutralization. N6 had a moderate binding affinity of $1.04 \times 10^8$ M against SARS-CoV-2 RBD (Fig 4B). Furthermore, N6 failed to block RBD binding to soluble ACE2 by ELISA (Fig 4C), suggesting it recognizes an epitope on RBD outside the ACE2 binding site.

While pseudoviruses are a good surrogate for identifying neutralizing antibodies, we next tested the ability of N6 to block infection by authentic SARS-CoV-2 virus using a cytopathic effect (CPE) based neutralization assay. The N6 mAb did not show significant reduction in neutralization potential in cytopathic effect–based assay when tested up to 20 μg/ml against live authentic SARS-CoV-2 live virus. The control purified Anti-RBD IgG from immunized mice sera showed neutralization CPE titers upto 0.1μg/ml. The II62 scFv-Fc antibody isolated from a semi-synthetic library [26] was used as experimental negative control (Fig 5).

## Polyclonal plasma from children with chronic HIV-1 infection neutralize multiple coronaviruses

We next tested plasma antibodies of children with chronic HIV-1 infection for their ability to bind SARS-CoV-2 S2P$_{ecto}$ protein and RBD. Ten children that had shown potent neutralization titre against a 12-virus global panel of HIV-1 isolates from previous studies in our lab were selected [27–29]. While all ten children showed significant binding to both S2P$_{ecto}$ and RBD (Fig 6A), three children showed potent and near-complete neutralization of SARS-CoV-2 pseudoviruses (AIIMS329, AIIMS330, AIIMS346) while two children (AIIMS355 and AIIMS521) showed moderate neutralization of SARS-CoV-2 (Fig 6B and 6C).

Antibodies generated against common endemic coronaviruses (HKU1, OC43 and 229E) have been shown to cross-react with SARS-CoV-2. To understand whether the SARS-CoV-2 neutralization titres observed in AIIMS329, AIIMS330, AIIMS346, AIIMS355 and AIIMS521 were due to HIV-1 bnAbs or antibodies to other coronaviruses contributed to neutralization, we next tested the neutralization of SARS-CoV-1, HKU1, OC43, MERS, NL63 and OC43 pseudoviruses. Of note, SARS-CoV-1, HKU1, OC43 and MERS belong to betacoronaviruses genus while NL63 and OC43 belong to alphacoronavirus genus. All five plasma samples that showed SARS-CoV-2 cross-neutralization potently neutralized HKU1, one of the viruses responsible for common cold while the extent of neutralization titres against OC43, NL63 and 229E were

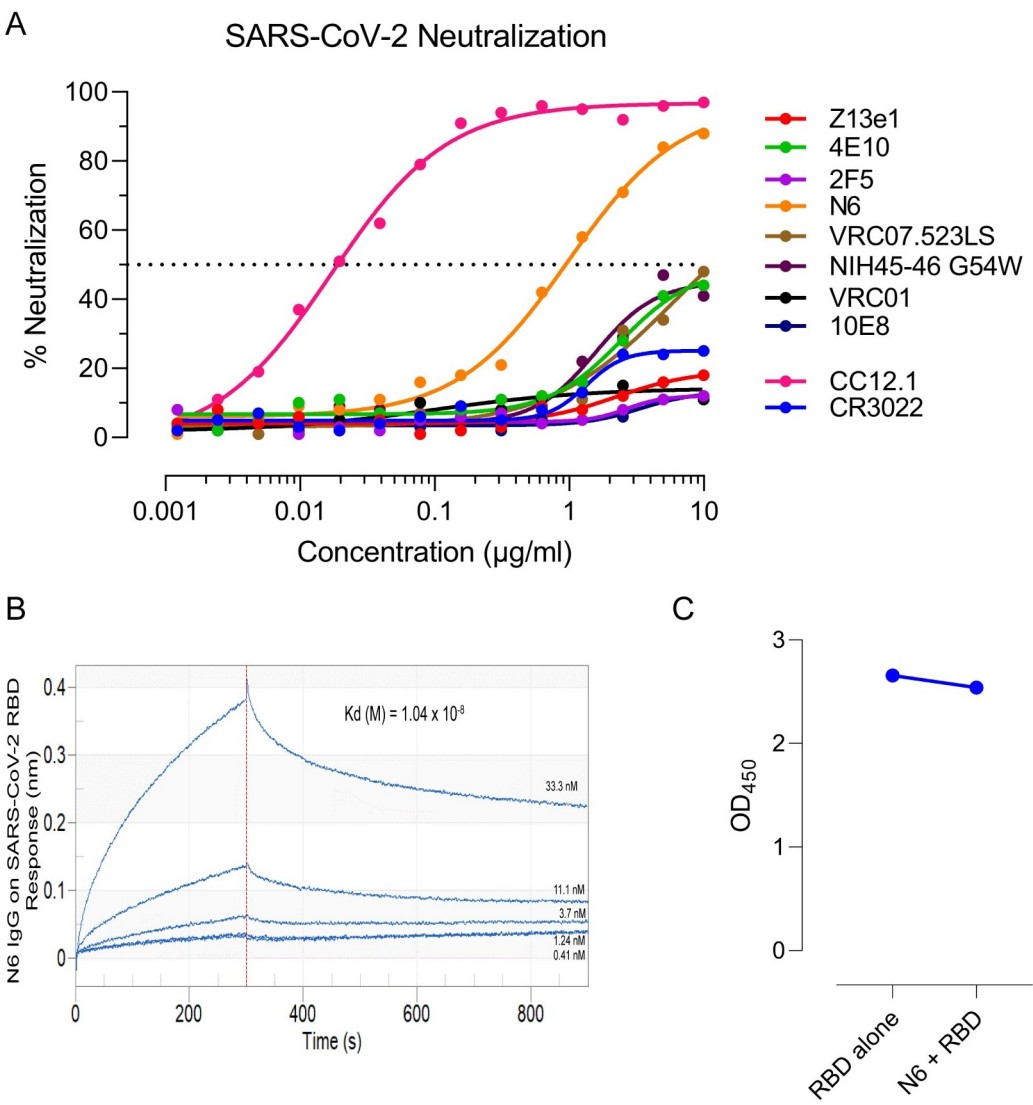

**Fig 4. Neutralization of SARS-CoV-2 by HIV-1 bnAbs.** (A) The bnAbs were tested for neutralization of pseudotyped SARS-CoV-2 virions. Percent neutralization was calculated by assessing relative luminescence units (RLU) in cell lysates of HEK293T-ACE2 cells 48 hours after infection with SARS-CoV-2 pseudoviruses in the presence of increasing amounts of bnAbs (range, 0.0048 to 10 μg/ml). N6, an anti-HIV-1 CD4-binding site bnAb, showed cross-neutralization of SARS-CoV-2. Dotted line corresponds to 50% neutralization. Graphs were plotted using average values (percent neutralization) from three independent experiments. (B) Affinity of N6 against SARS-CoV-2 RBD was measured using biolayer interferometry. (C) Competition ELISA was performed for RBD binding to ACE2 in presence and absence of N6. Average $OD_{450}$ value from three independent experiments are shown. CR3022, a SARS-CoV neutralizing antibody, and CC12.1, a SARS-CoV-2 neutralizing antibody, were used as positive control.

moderately weak (Fig 6D and 6E). None of the five samples showed any significant cross-neutralization of MERS while SARS-CoV-1 was neutralized by all five samples to the similar extent as SARS-CoV-2 (Fig 6D and 6E).

## Discussion

Herein, we show that broadly neutralizing antibodies targeting the envelope glycoprotein (gp160) of HIV-1 cross-react with the receptor binding domain (RBD) of SARS-CoV-2.

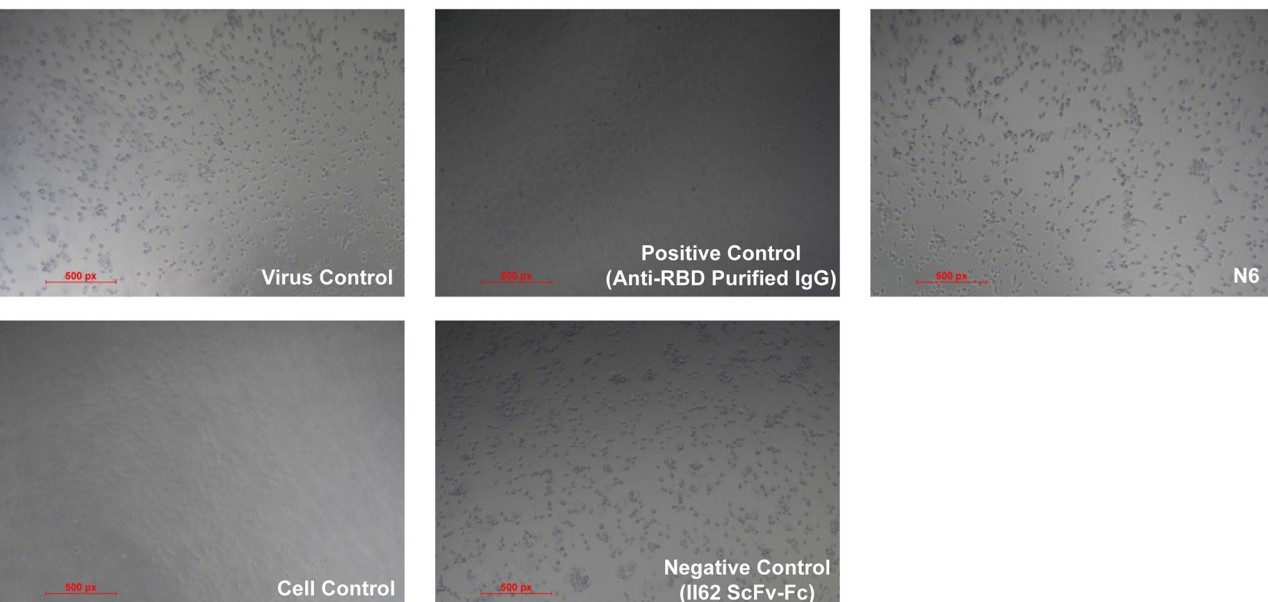

**Fig 5. N6 failed to neutralize authentic SARS-CoV-2 virus.** A) SARS-CoV-2 infected Vero-E6 cells monolayer after 72 hours post infection is shown as virus control B) Uninfected Vero-E6 cells monolayer after 72 hours showing complete absence of CPE is shown as representative cell control. Incubation of SARS-CoV-2 virus with 20 µg/ml of C) N6, D) II62 scFv-Fc and E) Anti-RBD IgG respectively, followed by adsorption for 60 minutes on Vero E6 cells. CPE was observed after 72 hours. No significant reduction in the CPE (neutralizing activity) was observed for N6 & II62 mAb which indicates these antibodies did not block SARS-CoV-2 infection. The purified Anti-RBD IgG was used as assay positive control and no CPE was observed (upto 0.1µg/ml) which means virus is completely blocked by neutralizing antibodies and did not show infection on Vero-E6 cells Scale, 500px.

Neutralizing antibodies engage the host immune system to clear the pathogen or infected cells and are promising candidates for combating emerging viruses [30–32]. The RBD of coronaviruses are highly immunogenic and infected individuals typically mount a nAb response [10,33–37]. Given that several HIV-1 bnAbs showed cross-reactivity with RBD of SARS-CoV-2, vaccine efforts should focus inducing antibodies targeting the cross-reactive epitopes on RBD. HIV-1 bnAbs have several atypical features that are generally not seen in antibody responses generated against a wide variety of pathogens. One such feature of HIV-1 antibodies is their poly- and/or autoreactivity which might have been the cause behind the cross-reactivity observed with SARS-CoV-2 RBD. Furthermore, Polyclonal antibody responses generated in chronic stages of HIV-1 tend to be poly- and/or autoreactive [38,39]. These features are not typically associated with antibody responses to most pathogens and are one of the major obstacles in HIV-1 bnAb-based therapies. HIV-1 bnAbs take years to develop, have atypical features including long complementarity-determining regions (CDR) loops, high levels of somatic hypermutations (SHMs), presence of insertions and/or deletions (indels), tyrosine sulfation, and develop to tolerate significant alterations in their core epitope. Furthermore, several of HIV-1 bnAbs have been shown to be poly and/or autoreactive. Autoreactive properties have been described for several HIV-1 bnAbs, including 2F5, 4E10, !0E8, VRC01, PGT121 and PGT151 [40–44]. Such bnAbs have been suggested to arise, in part, due to the breakdown in tolerance pathways that would otherwise delete and/or inactivate such virus-specific B-cells.

While several HIV-1 bnAbs showed cross-reactivity in binding assay, only N6, a member of the VRC01 class of HIV-1 bnAbs, was able to neutralize SARS-CoV-2 pseudovirus though it failed to neutralize authentic SARS-CoV-2 virus in a cytopathic effect-based neutralization assay. While pseudoviruses are a great tool to understand viral biology in a biosafe environment, one of the key limitations with pseudoviruses is the enhanced neutralization

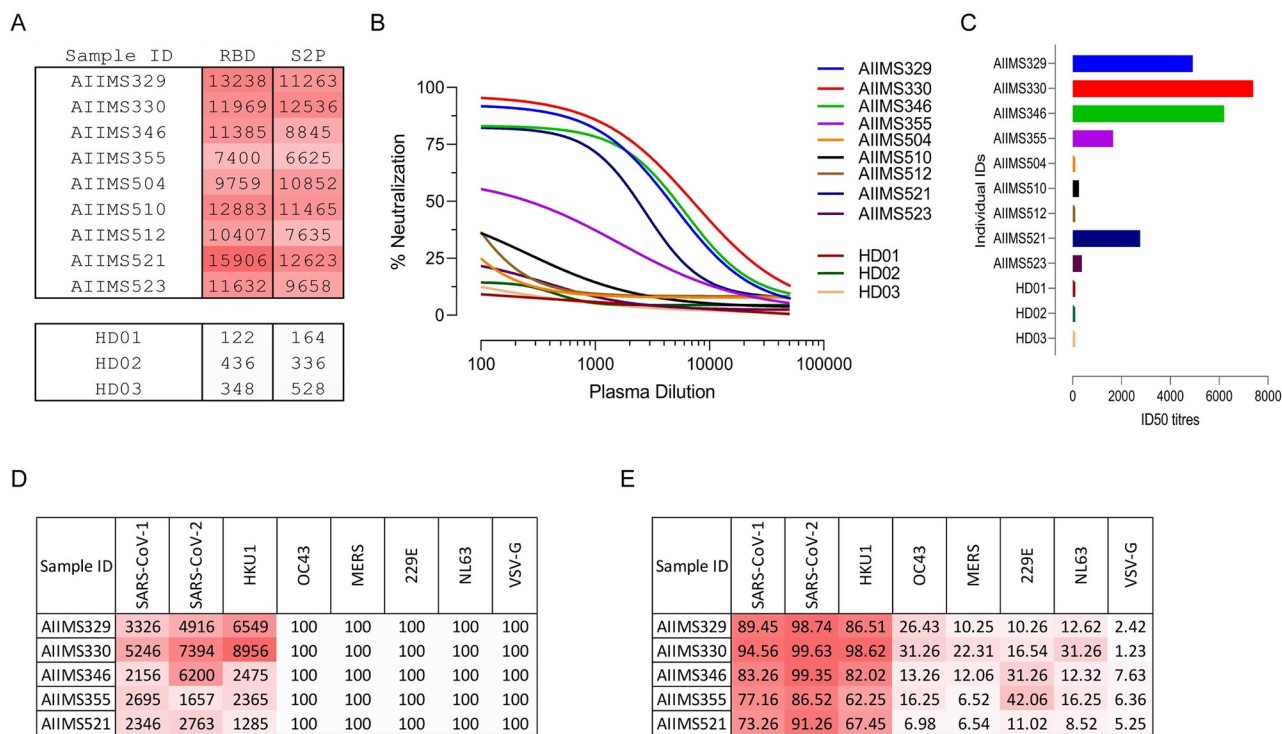

**Fig 6. Polyclonal plasma of HIV-1 infected children neutralizes SARS-CoV-2.** (A) Cross-reactivity of anti-HIV-1 neutralizing plasma antibodies from ten children with chronic HIV-1 infection against SARS-CoV-2$_{RBD}$ and SARS-CoV-2 S2P$_{ecto}$ was assessed by ELISA. Plasma antibodies from three seronegative healthy donors were used as negative control. Area under the curve (AUC) of OD$_{450}$ values of a 12-point binding curve (range, inverse plasma dilution of 100 to 51200), from three independent experiments are shown. (B) Plasma samples were tested for their neutralization of pseudotyped SARS-CoV-2 virions. Percent neutralization was calculated by assessing relative luminescence units (RLU) in cell lysates of HEK293T-ACE2 cells 48 hours after infection with SARS-CoV-2 pseudoviruses in the presence of increasing dilution of plasma samples (range, inverse plasma dilution of 100 to 51200). (C) Respective ID$_{50}$ (50% inhibitory dilution) for plasma from all ten children are shown. (D) ID50 titres against lentiviral pseudoviruses of all seven coronaviruses from all five children that cross-neutralized SARS-CoV-2 (range, inverse plasma dilution of 100 to 51200). (E) Maximum percent neutralization of all seven pseudotyped coronaviruses at the fixed plasma dilution of 1:100.

susceptibility. Several HIV-1 bnAbs have been shown to potently neutralize several HIV-1 pseudoviruses but fail to neutralize infectious molecular clones and/or primary isolates of HIV-1 [45–47]. It is noteworthy that N6 recognizes HIV-1 in an unusual orientation and neutralizes HIV-1 isolates that are typically resistant to other VRC01 class bnAbs [48]. Furthermore, it can tolerate absence of key CD4bs antibody contact residues across the length of heavy chain and can tolerate escape mutations that typically provide resistance to HIV-1 from other CD4bs bnAbs. Of note, N6 has an unprecedented degree of somatic hypermutation (31% in heavy and 25% in light chain at the nucleotide level).

One of the key limitations of the current study is the lack of a detailed structural analysis of cross-reactive bnAbs with SARS-CoV-2. A detailed structural analysis of N6 binding with SARS-CoV-2 will provide an insight of the exact nature of epitope-paratope interaction that contribute to the cross-reactivity observed herein. While the current work was designed to show the ability of polyreactive and somatically hypermutated antibodies generated in chronic HIV-1 infection to bind to new and emerging pathogens such as SARS-CoV-2, our results provide impetus for future detailed structural studies.

Most HIV-1 vaccine candidates are in the stage where they typically induce tier 1B or 2 responses against autologous and heterologous viruses in rabbits and non-human primates [49–51]. A germline targeting HIV-1 candidate immunogen (eOD-GT8) which was designed

to prime VRC01 class CD4bs directed antibodies has been described and the frequencies and affinity of B cells from healthy HIV-1 uninfected individuals recognizing this germline-targeting immunogen showed its suitability as a candidate human vaccine prime [52,53]. Naive B-cells that recognised eOD-GT8 had L-CDR3 sequences that matched several VRC01 class bnAbs, suggesting B-cells with light chain sequences for VRC01 class exist at high frequency. Based on the above observations and the availability of sera from these immunized animals, and findings herein of the ability of HIV-1 CD4bs directed bnAbs to inhibit SARS-CoV and SARS-CoV-2 pseudovirus infection, it is pertinent that the immune sera be tested for binding and neutralization of SARS-CoV-2. Furthermore, detailed structural studies should be taken with N6 to identify its epitope and neutralization determinants, which can be used to engineer its variants as effective SARS-CoV-2 therapeutics.

Collectively, our findings highlight the ability of HIV-1 specific bnAbs and polyclonal plasma to cross-react with the newly emerged SARS-CoV-2. Understanding cross-reactive neutralization epitopes of antibodies generated in divergent viral infections can provide key evidence for engineering so called super-antibodies (antibodies that can potently neutralize diverse pathogens with similar antigenic features). Recently, using a rapid affinity maturation strategy, Zhao et.al., engineered the CR3022, a neutralizing antibody against SARS-CoV-1, to potently neutralize SARS-CoV-2 and highlighted the potential of engineering approaches that can be leveraged to refocus an existing neutralizing antibody to target a related but resistant virus [54]. Furthermore, generating libraries using the starting sequence of such cross-reactive antibodies using approaches such as rapid affinity maturation, can provide candidate antibodies in a short frame of time upon which synthetic variants can be generated in the face of future pandemics.

## Methods

### Ethics statement

The study was approved by the institute ethics committee of All India Institute of Medical Sciences (IEC/NP-295/2011 & IEC/59/08.01.16).

### Study design

The current study was designed to assess the cross-reactivity of HIV-1 broadly neutralizing antibodies and plasma antibodies from children with chronic HIV-1 infection against the SARS-CoV-2. After written informed consent from guardians, blood was drawn in 3-ml EDTA vials, and plasma was aliquoted for plasma neutralization assays and binding ELISAs.

### Cell lines

HEK293T cells for pseudovirus production and generation of 293T-ACE2 cells, Vero-E6 cells, 293T-ACE2 and Huh7 cells for all seven coronavirus pseudovirus neutralization assays were maintained at 37˚C in 5% $CO_2$ DMEM containing 10% heat-inactivated FBS (vol/vol), 10mM HEPES, 1mM sodium pyruvate, and 100 U ml$^{-1}$ penicillin/streptomycin. Expi293F cells for recombinant antigen and monoclonal antibody production (Thermo Fisher Scientific, A1452) were maintained at 37˚C in 8% $CO_2$ in Expi293F expression medium (Thermo Fisher Scientific, A1435102).

### Plasmids

phCMV3 expression plasmids encoding the soluble S2P ectodomain of SARS-CoV (residue 1–1190), SARS-CoV-2 (residue 1–1208), RBD domain of SARS-CoV (residue 319–513),

SARS-CoV-2 RBD (residue 332–527), and full-length spikes of SARS-CoV-1, SARS-CoV-2, MERS, HKU1, OC43, NL63 and 229E were kindly gifted by Dr. Raiees Andrabi (The Scripps Research Institute). pCR3 expression vectors encoding truncated version of SARS-CoV S protein (residue 1–1236) and SARS-CoV-2 S protein (residue 1–1254), and pNL4-3ΔEnv-nanoluc were kindly gifted Dr. Paul Bieniasz (The Rockefeller University). CMV-Luc, RΔ8.2 backbone plasmid, pTMPRSS2 were kindly gifted by Dr. Barney Graham (National Institutes of Health). For positive controls in binding and neutralization assays, CR3022 and CC12.1 heavy and light chains were synthesized commercially and subcloned in phCMV3.

## Bacteria

E. coli DH5α, DH10β and STBL3 for propagation of plasmids were cultured at 37˚C (30˚C for STBL3) in LB broth (Sigma-Aldrich) with shaking at 220 rpm.

## Plasma from children with chronic HIV-1 infection

Well-characterized plasma sample from ten children that had shown potent neutralization titre against a 12-virus global panel of HIV-1 isolates from previous studies in our lab were selected.

## Recombinant protein production and purification

SARS-CoV and SARS-CoV-2 ectodomain and RBD constructs were transiently transfected in Expi293F cells at a density of 2 million cells/mL using polyethylenimine and expression plasmids at a molar ratio of 3:1 and purified from clarified transfected culture supernatants 4-days post-transfection using $Ni^{2+}$-NTA affinity chromatography (GE Life Sciences). Proteins were eluted from the column using 250 mmol/L imidazole, dialyzed into phosphate buffered saline (PBS), pH 7.2 and concentrated using Amicon 10-kDa (RBD) and 100-kDa ($S2P_{ecto}$) Amicon ultra-15 centrifugal filter units (EMD Millipore). Protein concentration was determined by the Nanodrop method using the protein molecular weight and molar extinction coefficient as determined by the online ExPASy software (ProtParam).

## Antibody production and purification

The monoclonal antibodies (PGT145, CAP256.25, VRC01, 10–1074, BG18, AIIMS-P01, and PGT151 for HIV-1, and CR3022 and CC12.1 for CoV-2) were expressed by co-transfection of heavy chain and light chain IgG1 plasmids (1:1 molar ratio) in Expi293F cells at a density of 0.8–1.2 million cells/mL using PEI-Max (1:3 molar ratio) as the transfection reagent. Five days post-transfection, antibodies were purified from clarified supernatants using protein A beads, eluted with IgG elution buffer and concentrated using 10-kDa Amicon ultra-15 centrifugal filter units (EMD Millipore).

## Binding ELISA

96-well microtiter plates were coated overnight with 2 µg/ml of purified SARS-CoV $S2P_{ecro}$, SARS-CoV-2 $S2P_{ecto}$, SARS-CoV RBD and SARS-CoV-2 RBD. Plates were blocked with 1% BSA for 3 hours. Monoclonal antibodies were added at a starting concentration of 10 µg/ml, with 11-point titration, and incubated for 2 hours at room temperature. Horseradish peroxidase conjugated goat anti-human IgG was used as secondary antibody and TMB substrate was used for color development. Absorbance at 450 nm was measured using a spectrophotometer. CR3022 (a nAb isolated from a convalescent SARS-CoV patient, which has been shown to cross-react with SARS-CoV-2), and CC12.1 (a SARS-CoV-2 neutralizing antibody isolated

from a convalescent donor) were used as positive control while two antibodies targeting the envelope glycoprotein of simian immunodeficiency virus (SIV) were used as negative control.

## Generation of 293T-ACE2 cells

VSV-G pseudotyped lentiviruses packaging the human ACE2 were generated by co-transfecting the HEK293T cells with pHAGE6-CMV-ACE2-ZsGreen plasmid and lentiviral helper plasmids (HDM-VSV-G, HDM-Hgpm2, HDM-Tat and CMV-Rev). 48 hours post-transfection, lentiviruses were harvested and used to infect HEK293T cells pre-seeded 24-hours in the presence of 10 μg/ml polybrene. 3-days post-infection, transduced cells were sorted via flow cytometry and maintained as a polyclonal pool of 293T-ACE2 cells in DMEM containing 10% heat-inactivated FBS (vol/vol), 10mM HEPES, 1mM sodium pyruvate, and 100 U $ml^{-1}$ penicillin/streptomycin at 37˚C in 5% $CO_2$.

## Viruses

To generate HIV-1 based pseudotyped viral stocks of all seven coronaviruses, HEK293T cells were co-transfected with CMV-Luc, RΔ8.2 backbone plasmid, and full-length spikes of SARS-CoV-$S_{trunc}$, SARS-CoV-$2_{trunc}$, MERS, HKU1, OC43, NL63 and 229E using polyethylenimine. Six hours post-transfection, cells were washed twice with RPMI and fresh media (10% DMEM) was added. Supernatants containing virions were harvested 72 hours post-transfection, filtered and stored at -80˚C. infectivity of pseudoviruses was determined by titration on 293T-ACE2 cells.

## Neutralization assays

Full length spike proteins of all seven coronaviruses were co-transfected with an HIV-1 backbone and helper plasmid expressing firefly luciferase and for SARS-CoV-1 and -2, serine protease TMPRSS2 (CMV-Luc, RΔ8.2 backbone plasmid, pTMPRSS2) in 1.25 x $10^5$ HEK293T cells for 72 hours. Post-transfection, culture supernatants were harvested, filtered and stored at -80˚C. For determination of neutralization potential of bnAbs against SARS-CoV-2, eight-point titration curves with 2-fold serial dilution starting at 10 μg/ml, were performed. For neutralization potential of pediatric plasmas with chronic HIV-1 infection against all seven coronaviruses, eight-point titration curve starting at plasma dilution of 1:100 and going upto 1:51200 were used. Serially diluted bnAbs and plasmas were mixed with respective pseudotyped viruses for 1 hour at 37˚C. pseudovirus/bnAb combinations were then added to 293T-ACE2 cells pre-seeded (24-hours) at 20,000 cells/well. For plasma samples, plasma/pseudovirus combination was added to Vero-E6 cells (20,000 cells/well). After 48–72 hours, supernatant was removed and luminescence was measured on Tecan luminescence plate reader using Bright Glow reagent. The percent infectivity was calculated as ratio of relative luminescence units (RLU) readout in the presence of bnAbs and plasmas normalized to RLU readout in the absence of mAb and plasmas. The half maximal inhibitory concentrations (IC50 for antibodies, ID50 for plasmas) were determined using 4-parameter logistic regression (GraphPad Prism version 8.3). CC12.1, a neutralizing antibody against SARS-CoV-2, was used as positive control. Amphotropic envelope protein from Vesicular Stomatitis Virus (VSV-G) and murine leukemia virus (MLV) were used as positive virus infection controls.

## Biolayer interferometry analysis of the SARS-CoV-2 RBD binding affinity with N6 bnAb

Biolayer interferometry was performed using an Octet Red96 instrument (ForteBio, Inc.). A 5 μg/ml concentration of SARS-CoV-2 RBD-His was immobilized on a Ni-NTA coated

biosensor surface. The baseline was obtained by measurements taken for 30 s in running buffer (1x PBS, 0.1% BSA and 0.02% Tween-20), and then, the sensors were subjected to association phase immersion for 300 s in wells containing N6 bnAb diluted in running buffer. Then, the sensors were immersed in running buffer for 600 s to measure dissociation. Biosensor was then regenerated by dipping it in EDTA followed by nickel sulfate solution. The mean $K_{on}$, $K_{off}$ and apparent $K_D$ values of the SARS-CoV-2 RBD binding affinity for N6 bnAb were calculated from all the binding curves based on their global fit to a 1:1 Langmuir binding model.

## Cytopathic effect (CPE) based SARS-CoV-2 neutralization assay

CPE-based neutralisation assays were carried out as previously described in Parray et al; 2020. In brief, $1 \times 10^2$ $TCID^{50}$ isolate USA-WA1/2020 virus was passaged once in Vero-E6 cells and then treated with antibody dilutions ranging from 20 to 0.3μg/ml for 90 minutes before adsorption on Vero-E6 cells for 1 hour. Following cell washing, DMEM supplemented with 2% FBS was added. After 4–5 days of incubation at 37˚C with 5% CO2, the presence of cytopathic effect (CPE) in cells was observed using a microscope. Non-infected Vero-E6 cells and purified IgG from RBD immunized mice were used as a positive control, and infected Vero-E6 cells and II62 scFv-Fc were used as a negative control.

## Cell surface binding assay

$1.25 \times 10^5$ HEK293T cells seeded in a 12-well plate were transiently transfected with 1.25 μg of SARS-CoV-2 S full-length protein using PEI-MAX. 48 hours post-transfection, cells were harvested and per experimental requirement, distributed in 1.5 ml microcentrifuge tubes. For monoclonal antibody staining, 10 μg/ml of antibody was used and titrated 2-fold in staining buffer. 100 μl of primary antibody (HIV-1 specific monoclonals) were added to HEK293T cells expressing SARS-CoV-2 S, and incubated for 30 minutes at room temperature. CR3022 and CC12.1 were used as positive control for surface binding assay. After washing, 100 μl of 1:500 diluted PE conjugated mouse anti-human IgG Fc was added, and after 30-minute incubation, a total of 50,000 cells were acquired on BD LSRFortessa X20. Data was analyzed using FlowJo software (version v10.6.1).

## Statistics and reproducibility

All statistical analyses were performed on GraphPad Prism 8.3. A p-value of $<0.05$ was considered significant. Neutralization assays were performed in triplicates and repeated thrice. Average IC50 values are shown and used for statistical comparisons. Binding ELISAs were performed in duplicates and repeated thrice. Average $OD_{450}$ values were used for plotting curves. Surface binding assay was performed thrice and average PE-MFI (phycoerythrin-median fluorescence intensity) values were used for plotting curves.

## Acknowledgments

We are grateful to Dr. Raiees Andrabi for providing the $S2P_{ecto}$ and RBD constructs for SARS-CoV and SARS-CoV-2, full length spike constructs of SARS-CoV-1, SARS-CoV-2, MERS, HKU1, OC43, 229E and NL63, Dr. Barney Graham for providing CMV-Luc, RΔ8.2 backbone plasmid, pTMPRSS2 and Dr. Paul Bieniasz for providing the full-length envelope constructs of SARS-CoV, SARS-CoV-2 and pNL4-3ΔEnv-nanoluc. We are thankful to NIH AIDS Reagent program for providing HIV-1 envelope pseudovirus plasmids, bnAbs, non-nAbs and their expression plasmids, and TZM-bl cells, and Neutralizing Antibody Consortium (NAC), IAVI,

USA for providing bnAbs. We are thankful to Dr. Michel Nussenzweig for providing 10–1074 and BG18 bnAb expression plasmids.

## Author Contributions

**Conceptualization:** Nitesh Mishra, Kalpana Luthra.

**Data curation:** Nitesh Mishra, Sanjeev Kumar, Rajesh Kumar, Sankar Bhattacharyya.

**Formal analysis:** Nitesh Mishra.

**Funding acquisition:** Kalpana Luthra.

**Investigation:** Nitesh Mishra, Sanjeev Kumar, Swarandeep Singh, Tanu Bansal, Nishkarsh Jain, Sumedha Saluja, Rajesh Kumar, Sankar Bhattacharyya.

**Methodology:** Nitesh Mishra, Sanjeev Kumar, Swarandeep Singh, Tanu Bansal, Nishkarsh Jain, Rajesh Kumar, Sankar Bhattacharyya, Jayanth Kumar Palanichamy, Riyaz Ahmad Mir, Subrata Sinha.

**Project administration:** Jayanth Kumar Palanichamy, Riyaz Ahmad Mir, Subrata Sinha, Kalpana Luthra.

**Resources:** Sanjeev Kumar, Sumedha Saluja, Jayanth Kumar Palanichamy, Riyaz Ahmad Mir, Subrata Sinha, Kalpana Luthra.

**Supervision:** Jayanth Kumar Palanichamy, Riyaz Ahmad Mir, Subrata Sinha, Kalpana Luthra.

**Visualization:** Nitesh Mishra.

**Writing – original draft:** Nitesh Mishra, Kalpana Luthra.

**Writing – review & editing:** Nitesh Mishra, Sanjeev Kumar, Swarandeep Singh, Tanu Bansal, Jayanth Kumar Palanichamy, Riyaz Ahmad Mir, Subrata Sinha, Kalpana Luthra.

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
