## [Decision Letter · Decision Letter 0]

15 Sep 2021

Dear Dr. Luthra,

We are pleased to inform you that your manuscript 'Cross-neutralization of SARS-CoV-2 by HIV-1 specific broadly neutralizing antibodies and polyclonal plasma' has been provisionally accepted for publication in PLOS Pathogens.

Best regards,

Kanta Subbarao

Section Editor

PLOS Pathogens

Kanta Subbarao

Section Editor

PLOS Pathogens

Kasturi Haldar

Editor-in-Chief

PLOS Pathogens

orcid.org/0000-0001-5065-158X

Michael Malim

Editor-in-Chief

PLOS Pathogens

orcid.org/0000-0002-7699-2064

I invited an additional expert to review your manuscript in addition to the Creative Commons review. The comments are attached.

Reviewer Comments (if any, and for reference):

Reviewer's Responses to Questions

**Part I - Summary**

Reviewer #1: I am satisfied with the response

**Part II – Major Issues: Key Experiments Required for Acceptance**

Reviewer #1: see above

**Part III – Minor Issues: Editorial and Data Presentation Modifications**

Reviewer #1: see above

PLOS authors have the option to publish the peer review history of their article (what does this mean?). If published, this will include your full peer review and any attached files.

Reviewer #1: **Yes: **Stephen Kent

---

## [Editor Report · Acceptance letter]

22 Sep 2021

Dear Dr. Luthra,

We are delighted to inform you that your manuscript, "Cross-neutralization of SARS-CoV-2 by HIV-1 specific broadly neutralizing antibodies and polyclonal plasma," has been formally accepted for publication in PLOS Pathogens.

Best regards,

Kasturi Haldar

Editor-in-Chief

PLOS Pathogens

orcid.org/0000-0001-5065-158X

Michael Malim

Editor-in-Chief

PLOS Pathogens

orcid.org/0000-0002-7699-2064